# How patients interpret early signs of foot problems and reasons for delays in care: Findings from interviews with patients who have undergone toe amputations

Alyson J. Littman[1,2,3]*, Jessica Young[2], Megan Moldestad[2], Chin-Lin Tseng[4], Joseph R. Czerniecki[5,6,7], Gregory J. Landry[8], Jeffrey Robbins[9], Edward J. Boyko[1,3,10], Michael P. Dillon[11]

1 Department of Veterans Affairs Puget Sound Health Care System, Seattle Epidemiologic Research and Information Center, Seattle, WA, United States of America, 2 Department of Veterans Affairs Puget Sound Health Care System, Seattle-Denver Center of Innovation for Veteran-Centered and Value-Driven Care, Seattle, WA, United States of America, 3 Department of Epidemiology, School of Public Health, University of Washington, Seattle, WA, United States of America, 4 Veterans Affairs New Jersey Healthcare System, East Orange, NJ, United States of America, 5 Department of Veterans Affairs Puget Sound Health Care System, Veterans Affairs Center for Limb Loss and Mobility (CLiMB), Seattle, WA, United States of America, 6 Department of Veterans Affairs Puget Sound Health Care System, Rehabilitation Care Services, Seattle, WA, United States of America, 7 Department of Rehabilitation, School of Medicine, University of Washington, Seattle, WA, United States of America, 8 Oregon Health & Science University, Portland, OR, United States of America, 9 Louis Stokes VA, Cleveland, OH, United States of America, 10 Department of Medicine, School of Medicine, University of Washington, Seattle, WA, United States of America, 11 Department of Physiotherapy, Discipline of Prosthetics and Orthotics, Podiatry, and Prosthetics and Orthotics, School of Allied Health, Human Services and Sports, La Trobe University, Melbourne, Victoria, Australia

* Alyson.littman@va.gov

## Abstract

### Aims

To describe how patients respond to early signs of foot problems and the factors that result in delays in care.

### Methods

Semi-structured interviews were conducted with a large sample of Veterans from across the United States with diabetes mellitus who had undergone a toe amputation. Data were analyzed using inductive content analysis.

### Results

We interviewed 61 male patients. Mean age was 66 years, 41% were married, and 37% had a high school education or less. The patient-level factors related to delayed care included: 1) not knowing something was wrong, 2) misinterpreting symptoms, 3) "sudden" and "unexpected" illness progression, and 4) competing priorities getting in the way of care-seeking. The system-level factors included: 5) asking patients to watch it, 6) difficulty getting the right type of care when needed, and 7) distance to care and other transportation barriers.

**Data Availability Statement:** Due to legal and ethical restrictions, we are unable to share data publicly because the data contain potentially

identifying and/or sensitive patient information. Subject to IRB approval, de-identified data will be released to a local VAPSHCS and/or national VA research data repository for release to non-VA protocols. The VA research data repository administrator will be responsible for reviewing and responding to requests to release data to non-VA requestors. A data use agreement compliant with VHA Handbooks 1200.12 and 1605.1 will be required between VHA and the requestor. Review and approval by VA privacy officer is required prior to disclosure. Data access requests will be reviewed by the IRB of the VA Puget Sound Health Care System (contact via Dr. Littman – Alyson. littman@va.gov), via mail address: 1660 S Columbian Way, Building 101 – 5W41, Seattle, WA 98108.

**Funding:** AJL, JY, MM, CLT were supported in part by funding from Veterans Health Administration, Office of Research and Development, Health Services Research & Development (IIR 15-372). The views expressed in this article are those of the authors and do not necessarily reflect the position or policy of the Department of Veterans Affairs or the United States government. The funders had no role in study design, data collection and analysis, decision to publish, or preparation of the manuscript.

**Competing interests:** The authors have declared that no competing interests exist.

## Conclusion

A confluence of patient factors (e.g., not examining their feet regularly or thoroughly and/or not acting quickly when they noticed something was wrong) and system factors (e.g., absence of a mechanism to support patient's appraisal of symptoms, lack of access to timely and convenient-located appointments) delayed care. Identifying patient- and system-level interventions that can shorten or eliminate care delays could help reduce rates of limb loss.

## Introduction

Amputations are a debilitating complication of diabetes. Between 10–25% of patients with diabetes develop a foot ulcer in their lifetime; of which approximately 5% will require an amputation [1]. The most common level of amputation among people with diabetes is the toe [2, 3]. These individuals are at high risk for subsequent amputations [4], largely because the factors that led to the initial amputation remain.

Many amputations in people with diabetes are thought to be preventable through education, self-management, and ongoing supervision by health care professionals [5]. To prevent amputations, patients must seek care soon after a problem is identified, because small wounds that are treated early are more likely to heal [6]. Untreated wounds are more likely to become infected and require an amputation to prevent spread to contiguous structures.

Prior studies of patients with major amputation and foot ulceration found that care delays were common [7–9]. Frequently-cited reasons for delays included misunderstandings about early warning signs and the seriousness of a break in the skin [7–9]. However, in these studies [7–9] patients with toe amputations were underrepresented, even though minor amputation rates exceed major amputation rates, and are on the rise [10–13]. Additionally, no prior studies focused on patients who receive care within the Department of Veterans Affairs (VA), the nation's largest integrated health care system. Lastly, none of these studies used a theoretical model to describe and understand delays in care. A framework [14–17] would help to identify points of intervention.

Because of VA's long-standing amputation prevention program that prioritizes patient education and self-care instruction [18], VA patients might better understand diabetes and foot-related complications compared with patients in the general population. However, VA patients may face other barriers to understanding because of their higher disability burden, lower income, and more frequently residing in rural areas.

Using a qualitative descriptive design, we aimed to describe how VA patients who had an initial toe amputation responded to early signs of foot problems and the factors that resulted in delays in care.

## Materials and methods

### Participants

We used medical records and referrals from providers to identify potential participants. We purposively sampled patients who met the inclusion criteria based on their medical records. The criteria included having an initial toe amputation no more than 12 months prior in VA or paid for by VA, a diabetes diagnosis, no prior history of lower extremity amputation (all based on medical records), and ability to ambulate prior to the toe amputation (based on self-report).

Details for the medical record-based inclusion/exclusion criteria are provided elsewhere [4]. To ensure a diversity of perspectives, we sought to include patients who had their initial toe amputation 1–2 months prior and 4–10 months prior as well as those who had undergone another amputation after the toe amputation.

## Recruitment

Each potentially eligible person was mailed a packet that included an information statement describing the study and a postage paid, pre-addressed postcard that could be returned to opt out of the study. If an opt out was not received, after providing the written information, study staff contacted potential participants by telephone to assess interest, answer questions, and, if the patient wished to participate, screen for eligibility, and obtain verbal informed consent. We obtained a waiver of the documentation of written informed consent. Verbal consent was confirmed at the start of the interview and recorded as part of documentation. During the screening call (prior to the interview), we confirmed that the toe amputation was their first amputation and that they were able to walk at least one city block prior to their amputation. Participants were offered $40 for completing an interview. Institutional review board approval was provided by VA Central IRB (#17–07).

## Data collection

The first two authors (AL and JY) developed a semi-structured interview guide. After the first few interviews we made minor modifications to clarify questions (Interview guide available in S1 File).

Two trained, master's level qualitative researchers (JY and MM) conducted the interviews via telephone between December 2017 and October 2018. Interviews began with broad, open ended questions, followed by specific follow-up questions and structured probes. Probes used participants' verbatim language to elicit thorough, detailed descriptions of events and patients' experiences prior to and following an amputation [19]. This inquiry, however, was limited to responses about events that preceded the toe amputation. Interviews were audio-recorded and transcribed verbatim.

We collected demographic and health information (e.g., marital status, education, and years since diabetes diagnosis) either at the end of the interview or at the time the interview was scheduled. Patient age was determined based on information in the VA medical record.

## Analysis

Data were analyzed concurrent with data collection using inductive content analysis, which involves open coding of data, organizing codes and associated data into categories, and com-paring data across participants to identify patterns/themes in the data [20]. During and imme-diately following each interview, interviewers wrote notes and summarized key aspects of the individual's experience, reflections on questions they asked or did not ask, and consistency and/or differences with prior hypotheses and interviews.

To become familiar with the data and create an initial codebook, the analysts (AL, JY and MM), reviewed interview transcripts and/or audio-recordings. The codebook was refined based on close review of the data and ongoing discussion among the analysts. Data were orga-nized and coded using ATLAS.ti qualitative software (version 8.4.15, Mnbh, Berlin). Coding discrepancies were resolved through discussion and consensus. Categories and themes were identified and refined through iterative data review, data comparison, and discussion.

Data collection and analysis continued until thematic saturation [21]. Analytic results, including themes and quotes, were reviewed by the larger investigative team (which included

clinicians and researchers with expertise in podiatry, epidemiology, qualitative research, internal medicine, vascular surgery, and physical medicine and rehabilitation) at multiple points to further refine themes. We ensured trustworthiness throughout study development, data collection, and analysis through frequent, reflexive memos and discussion with members within and outside the study team [22].

## Results

### Participants

We attempted to contact 341 patients but were unable to contact or screen 111 patients. Of the remainder, 114 were determined to be ineligible, 48 were not interested in participating, and 6 expressed an interest and were scheduled for an interview, but never participated (i.e., "no show"). We interviewed 62 patients. One person was found not to be eligible during the interview, leaving 61 completed interviews for analysis. Interviews lasted 45 minutes on average (SD = 12 minutes; range: 16–67 minutes). All participants were male; mean age was 66 years (Table 1). Thirty-seven percent had a high school degree or less and 13% had a college education or more. Nearly two-thirds (64%) had diabetes for more than 10 years. Two-thirds of participants had their toe amputation 4–12 months prior to the interview.

### Themes

We identified seven themes related to delayed care. We further subdivided themes depending on whether the factor was under the patient's control ("patient-level factors") or the system's control ("system-level factors"). As depicted in Fig 1, the patient-level factors included: 1) not knowing something was wrong, 2) misinterpreting symptoms, 3) having competing priorities getting in the way of care-seeking, and 4) "sudden" and "unexpected" infection progression;

**Table 1. Characteristics of patients with diabetes who had a first toe amputation in the year prior (n = 61).**

| Characteristic | N | % |
|---|---|---|
| **Age (years)** | | |
| <50 | 2 | 3 |
| 50–64 | 23 | 38 |
| 65–74 | 27 | 46 |
| 75+ | 9 | 15 |
| **Currently married** | | |
| No | 36 | 59 |
| Yes | 25 | 41 |
| **Education*** | | |
| High school graduate or less | 22 | 37 |
| Some college/Associate's (2 yr) degree | 30 | 50 |
| College graduate or more | 8 | 13 |
| **Years since first diagnosed with diabetes*** | | |
| <5 | 3 | 5 |
| 5–10 | 18 | 30 |
| 10+ | 39 | 64 |
| **Months since toe amputation** | | |
| 1–2 | 20 | 33 |
| 4–12 | 41 | 67 |

* Education and years since first diagnosed with diabetes missing for 1 person.

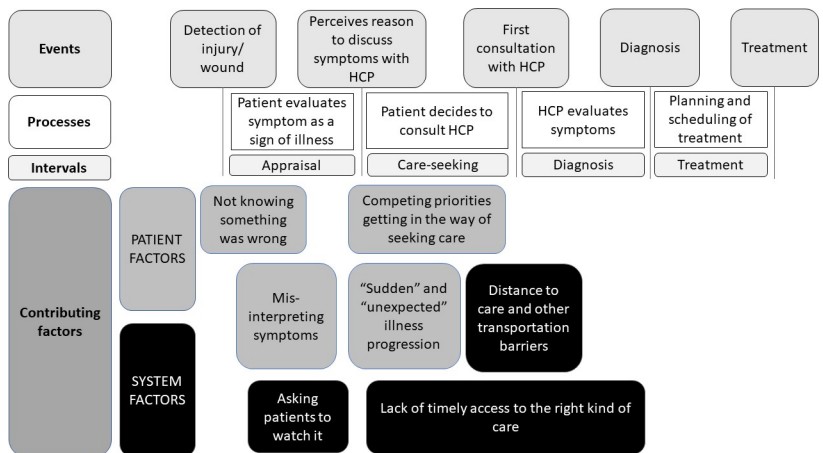

**Fig 1. Depiction of the stages of delay and how themes identified fit into those stages.** This figure was adapted from Walter et al. [15] and conceptualizes delay intervals occurring between phases of decision-making. The appraisal interval describes the time from becoming aware of bodily change to perceiving a reason to discuss symptoms with a healthcare provider (HCP). During this appraisal interval, patients may self-manage their foot. The care-seeking interval is the time between perceiving a reason to discuss symptoms with a HCP and meeting with a HCP. The diagnosis interval is the time between meeting with a HCP and having a provider diagnose the foot as having an ulcer/gangrene/etc. The treatment interval is the time between when a diagnosis is made until treatment is initiated. Two of the patient factors identified occurred during the appraisal interval, while two others occurred during the care-seeking interval. "Asking patients to watch it", a system-level factors impacted the appraisal interval because responsibility of evaluating worsening of symptoms was returned to the patient. Lack of timely access to care delayed care-seeking and diagnosis. Distance to care and other transportation barriers delayed care-seeking.

the system-level factors included: 5) asking patients to watch it, 6) lack of timely access to the right kind of care, and 7) distance to care and other transportation barriers. We provide details and exemplary quotes for each of these themes below; patient identifiers are listed in parentheses following the quotes.

## Patient-level factors

**Not knowing something was wrong.** Some patients reported not seeking care because they were unaware there was a problem. The most common reasons for not recognizing a problem was because they could not feel their feet and did not inspect their feet in a manner typically recommended for those with diabetes (e.g., on a daily basis, checking between toes, etc.). Some patients reported knowing that they should check their feet and having received education about foot checking; others reported not having received such instruction. Those who reported being told to inspect their feet followed this guidance in varying ways. Some did not check, some occasionally did a cursory look ["*I'd just basically look at my feet and say, 'Ok. They're good.'*"(00047)], and some said they looked as instructed.

When asked about checking his feet, this patient explained his process and what his feet felt like.

*I'm probably not the best at it. I hardly ever take my socks off, . . . It might be 3 or 4 days that I don't check. Because they feel like they're embedded in concrete anyway. The good side is I don't feel anything. The bad side is that's what caused it, getting an injury without knowing it. (3586930)*

For those who lacked sensation and did not check their feet, wounds were not apparent until the symptoms were glaring to either the patient or someone else (e.g., a friend, family

member, or clinician). These signs included a foul-smelling odor, a hole in the foot, massive swelling, and/or foot discoloration. As one patient explained,

*I had gangrene on my little toe. . . I didn't know it because I didn't feel it. My grandson. . . said, "Grandpa, there is a big hole in your foot." And I said, 'oh, BS.' And then I felt it and there was a big hole in my foot, and it was very putrid. I had a doctor's appointment, and I went in a couple days later, and they put me in the hospital and cut my toe off.* (01084).

The following is another example of a patient who had an infection to bone but was not aware of it until his doctor told him:

*I went in to have my toenails clipped. They noticed the callous on the bottom of my toe, and they would kind of trim it up a bit the couple of times I went. But the next time I went in the toe was real swollen. They got a doctor in there to check it and that's when he took the pus out of it. They sent me to the x-ray and then found out it got infected down to the bone. (00954)*

**Misinterpreting symptoms.** Some patients noticed symptoms but did not recognize them as an indication of a problem that required medical attention. Some patients explained that they regularly checked their feet but did not know how to identify early signs of foot problems. Early symptoms such as redness, swelling, or bruising were not evaluated as being a cause for concern or a reason to seek medical care. For example, one patient noticed something was different, but did not understand that it might mean he had an infection. He explained,

*I wish I would've known more about what to watch for in line with an infection. Because I didn't know what was going on with my toe, I never thought of there being an infection because there was no wound. It just started turning bruised like, and red. And turning sore.* (3571682).

Another patient thought that only in extreme cases required urgent medical care. When asked to explain what he thought he should be looking for, he responded,

*Looking to make sure there's not bleeding or anything crazy like that. . .any bad swelling, things like that. . .that's the things I thought meant I should go to the ER [Emergency Room].* (12860)

A few patients cited confidence in managing their symptoms because of prior medical training. However, even among patients without medical training, some thought it was something that they could handle on their own. For example, because basic first aid (e.g., topical antibiotics and a band aid) had worked in the past, they thought it would in this case, too.

*Sure, we {wife and patient} did what we always do–cleaned it, put some [antibiotic] cream on it, and wrapped or taped it up. . .whatever we've got to do to keep it from getting any worse or whatever. . . it always worked before.* (00404)

**Competing priorities getting in the way of seeking care.** Some patients noticed symptoms and thought it might require professional medical care but described delaying care because they needed to attend to family, work, or other health problems. For example, one patient explained that, prior to his toe amputation, he was helping his teenage son through a medical emergency.

*At the time, I didn't want to make a big deal out of it [his toes]. My . . . son was going through cancer treatments. . . I was going back and forth with him to the doctor, and I was focused on that. I wasn't about to try to take the focus off of that.* (3759990).

Another patient could not miss job trainings and thus missed his medical appointments, explaining,

*A bigger reason why I thought I lost my toe, is because I had started working and it was under [department], they found me this training thing to do. I only had a couple months of entitlements left, and my counselor. . .said that if I didn't miss any days, she'd give me a two month extension on my entitlements. But, she said, 'I don't think you can do it, I don't think you're ready to work, you're not motivated'. . ..I proved her wrong, but I lost my toe doing it; I didn't miss any days, but I missed a lot of appointments.* (12860)

For some patients, other medical problems impaired their ability to notice symptoms or to consider them important enough to address.

*Sometimes my medicine wasn't quite right. I was getting dizzy a lot. There was a lot going on with me at that time. I mean a whole bunch. And I was sick. Because I was septic. It was just a whole gamut of stuff. And I probably wasn't really paying attention to my foot like I should.* (11440)

**"Sudden" and "unexpected" infection progression.**   Some patients expressed surprise that what seemed like a minor cut developed "suddenly" and/or "unexpectedly" into a limb-threatening infection.

*It happened so fast, that it got infected . . . the toe looked like a cooked hotdog, that's how red it was. But it only took 2 days to get there, that's how fast it was.* (6227738)

Whether wounds or foot issues were new or persistent, patients did not see themselves as having the time or opportunity to seek earlier care because the problems worsened so quickly.

*Like I said, from the time it started turning grey, to the time that I had it amputated, we're talking 2 weeks, less than 2 weeks' time. And at that point, it was turning black. And I'm like, 'wow, it's just unbelievable that it went that fast.'* (14048)

Others described having persistent, stable, long-standing foot problems that seemed to suddenly and unexpectedly worsen. "*It's still really an unexpected thing, for this to happen to me . . . it was very fast.*" (7987093)

## System-level factors

**Asking patients to "watch" it.**   Some delays occurred after the patient obtained an appointment with a health care professional because they had concerns about symptoms. In these cases, delays occurred because the health care provider advised the patient to "watch it" and return if things worsened. This responsibility was problematic because many patients were uncertain about what they needed to watch for and how much "worse" it needed to be to seek care the next time. For example, one patient went to his podiatrist because he noticed that two of his toenails were turning gray. The provider advised him to watch his toes until his next appointment (which was scheduled for the following week) and to go to the emergency room if his symptoms worsened. He explained,

*She looked at it, and she wasn't sure what it was. . .she said to come back a week later, if it gets worse. . .she said to keep an eye on it. Well how am I supposed to keep an eye on it when I don't know what I am looking for?* (07118)

When the patient returned a week later for his appointment, he had gangrene and was told that he needed an amputation.

Another patient noticed a sore on his toe that did not seem to be getting better and made an appointment to see his primary care provider for an evaluation. The provider did not seem to think any action was needed and indicated that the patient should continue with self-treatment without changes.

*We went in, I took my socks and shoes off, and the bandage off. We told them what we were doing, that we were doctoring it, and he said, 'it looks good, keep up what you're doing, no problems.' So we came back home and we kept on doing what we were doing; my wife would put peroxide on it, then she'd put bacitracin on it, and then wrap it.* (3763534)

There was no follow-up planned and the toe did not improve. Within a week of the visit with the primary care provider, the toenail came off along with some skin, so the patient went to the emergency room and the provider said that they needed to amputate.

Other patients reported a different response from clinicians that included communication of concern, conducting testing to ascertain the extent of the problem/acuity, providing referrals as necessary, making and communicating a plan of care, and scheduling follow up appointments to check progress and determine next steps.

*Just, Dr. [podiatrist], she's right on top of everything. When this wound opened up, I went right into the office. She's very knowledgeable, very informative. . . She doesn't pull any punches. . . [she] told me 'we've got to get on top of this right away' and she did. . .went right in for testing and when that came back said, 'we've got a big problem here and we can't mess around. . ..we need to do this' [amputate].* (03722)

**Lack of timely access to the right kind of care.** When patients tried to seek care for foot problems, some recounted barriers to receiving appropriate and timely care. Many reported being unable to be seen quickly by a provider who was familiar with diabetic foot problems (e.g., a podiatrist, wound specialist, or vascular surgeon) and described calling VA for an appointment only to be told that they would need to wait weeks to be seen.

*That's about how long it takes to get in to see a doctor, 6 weeks to 2 months. . . And when he finally sent me down to the hospital, the surgeon, it again took almost 2 months. So, a lot of time went by just waiting to go to the doctor.* (3568976)

Sometimes patients had delays at multiple points–from the patient waiting to seek medical care to obtaining an appointment for an initial evaluation and then further testing to make a definitive diagnosis and treatment plan.

*In January I noticed a little slight swelling of my toes and stuff. . . It went on through January and in February I called the VA and talked to them and explained that I . . . noticed a dark spot show up on my big toe. At that time, we had to schedule with Prosthetics, which had moved from the main office in the hospital to an outside office, so there was a slight delay there before I could get in for an appointment . . . When I got there, the doctor looked over*

*everything and wanted to schedule me for x-rays and stuff in the main hospital, but we had to wait a couple of weeks prior to getting in due to the fact that they were so booked up.* (3670569)

The primary option for patients to be seen quickly was going to the Emergency Room, because of a lack of walk-in or same-day appointments. One patient explained,

*I originally tried to go to Podiatry directly. But I couldn't get in. If I waited for an appointment, I'd still be waiting. And then I said that I couldn't let this sit over the weekend, so from there I went to the ER . . . The young lady looked it up on the computer and at the time, the earliest appointment they [Podiatry] had available was like a month out. I knew I couldn't wait that long. So I said let's go with plan B, and that's when I went to the ER.* (787440)

**Distance to care and other transportation barriers.** Many patients explained that they lived a long distance from a clinic with diabetic foot care specialists, which was a barrier to receiving regular preventive care and having new concerns evaluated. For one patient, it was both the distance and his other health issues that discouraged him from going to the VA, unless it was absolutely necessary.

*Interviewer: Why not go in earlier?*

*Participant: I just, it's a long way for me to go to the VA. The clinic is 40 miles. And it's a pain in the ass. (11440)*

Other health concerns or comorbidities also impacted some patients' ability to travel long distances to receive foot care in a timely manner. For some patients, there were options closer to home. However, there were varied reasons why these other options were not feasible (e.g., because they lacked specialists or appointment times, or the VA would not grant them permission to get non-VA care paid for by VA—so they were forced to drive to clinics that were farther away).

Sometimes patients did not have their own transportation and had to rely on VA-provided transportation to take them to and from appointments. The VA transportation involved early pick-ups and late drop-offs, to accommodate all passengers attending their appointments. Multiple patients explained that taking transportation to the regional VA for a short appointment could take all day and was a deterrent to receiving care, with one patient saying it "*kills a whole day the way they schedule the appointments*" (3586930).

Another patient had a similar situation, explaining:

*Participant: Yeah. And I'd go back every other week at that time. That was hard on me.*

*Interviewer: Can you say more about how it was hard on you?*

*Participant: Because of going to [the regional VA], it was a 2-hour trip every day. And 2 hours coming back. That was the hardest part. In a van, I had to go up in a van.* (01484)

## Discussion

In interviews with a large sample of male patients from across the United States with diabetes who had undergone a toe amputation, delays in care were common. The delays included patients not noticing symptoms of an injury/infection, not identifying the symptoms as a sign of illness, and then being hindered in quickly seeing a healthcare provider. Many patients

perceived having little time to act more quickly because of how suddenly the wound seemed to progress, perhaps partly because early signs were not apparent to the patient. Delays in care also occurred after the patient sought care–because they were counseled to manage it themselves, were unable to obtain a timely appointment with a provider knowledgeable about the diabetic foot, and/or because it was difficult and/or time-consuming to get to clinics.

Many patients did not seem to understand the signs of ulceration and/or infection, nor the gravity of their symptoms; these issues have been identified in prior studies [7, 8, 23–28]. Few patients in our study had a college education, which may have impacted their understanding of diabetes. Nevertheless, other studies in people with diabetes both with [27] and without [28] a history of amputation or diabetic foot ulcers also found that patients had limited understanding about the connection between diabetes, glycemic control, and foot complications or how rapidly problems can progress [7]. Though education is emphasized in PAVE [18] and other programs, there is not strong evidence that structured patient education reduces the risk of ulceration or amputation [29, 30]. The reasons are not well understood, but recent research found that one-on-one education, which is the key modality used throughout the VA, was ineffective for most patients [31]. Specifically, for 58% of patients, there was no agreement between the podiatrists' key messages and the patients' explanation of those key messages immediately after the appointment [31]. The International Working Group of the Diabetic Foot recently concluded that instead of focusing solely on education, future studies should take a broader behavioral perspective, include different forms of structured education, and tailor the education based on patient behaviors and patient preferences [30].

Some patients cited competing priorities that led them to ignore early signs of foot problems and delay seeking care. Patients were caring for children and/or parents. Some had other health problems such as cardiovascular events that seemed more serious than their toes. Past experience–long waits to obtain an appointment, long drives to get to a clinic, and beliefs that symptoms would resolve with basic first aid—impacted care seeking. Opportunity costs–the time/effort it would take to get to a clinic–appeared to delay care. Consistent with Fortney et al.'s access to care model [32], past experience accessing care (e.g., expecting that they would have to wait 6–8 weeks for an appointment) influenced patients' perceptions of access and care seeking. Thus, to change patients' behaviors, it may first be necessary to change their perceptions, at least in part by the system becoming more responsive and care being more accessible.

Several factors should be considered when interpreting our results. First, our study was limited to Veterans and did not include any women, which could limit the generalizability of findings. The similarity between our results and those observed in non-Veteran and mixed gender populations suggest that our findings are not unique to Veterans or males [7, 8, 23–28]. We did not conduct member checking to ensure our understanding of patients' experiences was consistent with their own. We triangulated findings among three authors, who all read and/or listened to each interview. This process helped to ensure that the data analysis was a trustworthy representation of the themes rather than the reflection of one researcher's interpretations/biases [33]. Other strategies to enhance analytic rigor included going back and forth between the interview data, codes, categories, and themes to refine themes, ensure findings were grounded in the data, and validate results [33, 34]. We did not quantify the frequency that different themes arose because of the purposive and convenience sampling and our aim to understand the breadth of experience. Future studies utilizing quantitative methods may be better equipped to determine this.

Findings from this study suggest several opportunities to reduce delays in care and, consequently the risk of amputation. Identifying ways to make it easier for patients to know when something is wrong (e.g., that a foot lesion has developed or progressed) and then to act on that information is critical. Theory-based, evidence-based, and person-based approaches to

developing interventions to address gaps and barriers seem promising [23]. Approaches that use technology such as remote monitoring devices that could seamlessly share information about the patient's status with their healthcare providers, along with a protocol for acting on that information—could help address patients' challenges in knowing how to identify problems and accessing care when problems arise. Open access models and/or non-face-to-face contact–e.g., phone or video conference appointments–could also reduce barriers to timely access. These innovations could have enormous benefits in terms of averting limb loss and the many financial, personal, social, and societal costs that limb loss entails.

In summary, misunderstandings among diabetic patients undergoing a toe amputation allowed wounds to go unnoticed and/or untreated. Because there were barriers to accessing care, when care was sought, patient were unable to get the care they needed quickly enough to avoid an amputation. Fortunately, there are a growing number of options (e.g., telehealth, video appointments, secure patient-provider email, and remote monitoring) that may reduce delays in care, and consequently, limb loss rates.

## Supporting information

**S1 File. Interview guide.**
(DOCX)

## Acknowledgments

We thank Carolyn Klassen for her help recruiting participants. We also appreciate all the Veterans who shared their experiences with us.

## Author Contributions

**Conceptualization:** Alyson J. Littman.

**Data curation:** Alyson J. Littman, Jessica Young, Megan Moldestad.

**Formal analysis:** Alyson J. Littman, Jessica Young, Megan Moldestad.

**Funding acquisition:** Alyson J. Littman, Chin-Lin Tseng, Joseph R. Czerniecki, Gregory J. Landry, Jeffrey Robbins, Edward J. Boyko.

**Investigation:** Jessica Young.

**Methodology:** Alyson J. Littman, Jessica Young, Megan Moldestad.

**Project administration:** Alyson J. Littman.

**Supervision:** Alyson J. Littman.

**Validation:** Alyson J. Littman, Jessica Young, Megan Moldestad.

**Writing – original draft:** Alyson J. Littman.

**Writing – review & editing:** Alyson J. Littman, Jessica Young, Megan Moldestad, Chin-Lin Tseng, Joseph R. Czerniecki, Gregory J. Landry, Jeffrey Robbins, Edward J. Boyko, Michael P. Dillon.

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
