## [Decision Letter · Decision Letter 0]

13 Jan 2021

PONE-D-20-36785

How patients interpret early signs of foot problems and reasons for delays in care: findings from interviews with patients who have undergone toe amputations

PLOS ONE

Dear Dr. Littman,

Thank you for submitting your manuscript to PLOS ONE. After careful consideration, we feel that it has merit but does not fully meet PLOS ONE’s publication criteria as it currently stands. Therefore, we invite you to submit a revised version of the manuscript that addresses the points raised during the review process.

We look forward to receiving your revised manuscript.

Kind regards,

Arezoo Eshraghi, Ph.D.

Academic Editor

PLOS ONE

Journal Requirements:

2. In the Methods section, please provide a copy of the interview guides as Supporting Information.

Furthermore, When reporting the results of qualitative research, we suggest consulting the COREQ guidelines: http://intqhc.oxfordjournals.org/content/19/6/349. In this case, please provide additional information on how participants were approached for the interview, and where the interviews took place.

Reviewers' comments:

Reviewer's Responses to Questions

**Comments to the Author**

1. Is the manuscript technically sound, and do the data support the conclusions?

Reviewer #1: Yes

Reviewer #2: Yes

2. Has the statistical analysis been performed appropriately and rigorously? 

Reviewer #1: N/A

Reviewer #2: Yes

3. Have the authors made all data underlying the findings in their manuscript fully available?

Reviewer #1: Yes

Reviewer #2: Yes

4. Is the manuscript presented in an intelligible fashion and written in standard English?

Reviewer #1: Yes

Reviewer #2: Yes

5. Review Comments to the Author

Reviewer #1: Interesting qualitative study that was well designed.

Generalizability is always an issue for these types of studies. As is common for this type of research a large number of potential subjects were not ultimately participants. Only men were interviewed. Need to be careful about the implication of using the term “large national sample”. While it is, this term usually implies that the sample is representative, and this is rarely true for a qualitative study including this one.

It is interesting that patient understanding their medical condition was an issue for patients who are receiving education. This has been a common issue for educational programs and likely means we need to do a better job by maybe doing a “different” job.

Reviewer #2: Very well presented! It is rare that the authors have paid attention to the details of manuscript preparation. Thank you for your work and growing the edge of knowledge with your work. I look forward to seeing this in print!

Just an edit:

Pg5 line 115 do you mean "inquiry" rather than "analysis"

A pleasure to read!

6. PLOS authors have the option to publish the peer review history of their article (what does this mean?). If published, this will include your full peer review and any attached files.

Reviewer #1: No

Reviewer #2: No

---

## [Author Response · Author response to Decision Letter 0]

8 Feb 2021

** Response: We have re-reviewed the style requirements and done our best to abide by them.

2. In the Methods section, please provide a copy of the interview guides as Supporting Information.

** Response: We inadvertently failed to include the Supplemental File with the first submission. We apologize. The interview guide is included as S1 File.

Furthermore, When reporting the results of qualitative research, we suggest consulting the COREQ guidelines: http://intqhc.oxfordjournals.org/content/19/6/349. In this case, please provide additional information on how participants were approached for the interview, and where the interviews took place.

** Response: We appreciate the suggestion. We consulted the COREQ guidelines to ensure that we included the key information and presented it as clearly as possible; we made changes where we noticed opportunities to clarify our methods. We added a new subheading “Recruitment” where we detail how patients were approached. We state on line 127 that interviews were conducted over the telephone. 

** Response: We now specify that we obtained verbal consent and explain that we obtained a waiver of documentation of written informed consent in lines 114-115. Consent was verified at the start of the interview and recorded.

 ** Response: Due to legal and ethical restrictions, we are unable to share data publicly because the data contain potentially identifying and/or sensitive patient information. Data access requests will be reviewed by the IRB of the VA Puget Sound Health Care System (contact via Dr. Littman).

Reviewer #1: Interesting qualitative study that was well designed.

Generalizability is always an issue for these types of studies. As is common for this type of research a large number of potential subjects were not ultimately participants. Only men were interviewed. Need to be careful about the implication of using the term “large national sample”. While it is, this term usually implies that the sample is representative, and this is rarely true for a qualitative study including this one.

** Response: The reviewer makes a good point that our sample may not be representative. In line 405, we now state, “In interviews with a large sample of male patients from across the United States with diabetes who had undergone a toe amputation, delays in care were common.” We also include generalizability as a potential limitation in lines 444-445.

It is interesting that patient understanding their medical condition was an issue for patients who are receiving education. This has been a common issue for educational programs and likely means we need to do a better job by maybe doing a “different” job.

** Response: We agree that it is interesting that patients understanding their medical condition was an issue for patients who (we believe) are receiving education. As noted by the reviewer, this has been a common issue for education programs and suggests that more research is needed to determine how to make educational programs more effective.

Reviewer #2: Very well presented! It is rare that the authors have paid attention to the details of manuscript preparation. Thank you for your work and growing the edge of knowledge with your work. I look forward to seeing this in print!

Just an edit:

Pg5 line 115 do you mean "inquiry" rather than "analysis"

A pleasure to read!

** Response: We are pleased that you appreciated our paper. In prior line 115 (current line 130), we have replaced the word “analysis” with “inquiry.”

---

## [Editor Report · Decision Letter 1]

24 Feb 2021

How patients interpret early signs of foot problems and reasons for delays in care: findings from interviews with patients who have undergone toe amputations

PONE-D-20-36785R1

Dear Dr. Littman,

We’re pleased to inform you that your manuscript has been judged scientifically suitable for publication and will be formally accepted for publication once it meets all outstanding technical requirements.

Kind regards,

Arezoo Eshraghi, Ph.D.

Academic Editor

PLOS ONE
---

## [Editor Report · Acceptance letter]

2 Mar 2021

PONE-D-20-36785R1 

How patients interpret early signs of foot problems and reasons for delays in care: findings from interviews with patients who have undergone toe amputations 

Dear Dr. Littman:

I'm pleased to inform you that your manuscript has been deemed suitable for publication in PLOS ONE. Congratulations! Your manuscript is now with our production department. 

Kind regards, 

on behalf of

Dr. Arezoo Eshraghi 

Academic Editor

PLOS ONE